

# Combination of ferric ammonium citrate with cytokines involved in apoptosis and insulin secretion of human pancreatic beta cells related to diabetes in thalassemia

Patchara Rattanaporn[1,2,*], Sissades Tongsima[3,4],
Thomas Mandrup-Poulsen[5], Saovaros Svasti[2,6] and Dalina Tanyong[1,*]

[1] Department of Clinical Microscopic, Faculty of Medical Technology, Mahidol University, Nakhon Pathom, Thailand
[2] Thalassemia Research Center, Institute of Molecular Biosciences, Mahidol University, Nakhon Pathom, Thailand
[3] National Biobank of Thailand, National Science and Technology Development Agency, Pathum Thani, Thailand
[4] National Center for Genetics Engineering and Biotechnology, National Science and Technology Development Agency, Pathum Thani, Thailand
[5] Department of Biomedical Sciences, Faculty of Health and Medical Sciences, University of Copenhagen, Copenhagen, Denmark
[6] Department of Biochemistry, Faculty of Science, Mahidol University, Bangkok, Thailand
* These authors contributed equally to this work.

Corresponding author
Dalina Tanyong,
dalina.itc@mahidol.ac.th

## ABSTRACT

**Background:** Diabetes mellitus (DM) is a common complication found in β-thalassemia patients. The mechanism of DM in β-thalassemia patients is still unclear, but it could be from an iron overload and increase of some cytokines, such as interleukin1-β (IL-1β) and tumor necrosis factor-α (TNF-α). The objective of this study was to study the effect of interaction between ferric ammonium citrate (FAC) and cytokines, IL-1β and TNF-α, on 1.1B4 human pancreatic β-cell line.
**Methods:** The effect of the combination of FAC and cytokines on cell viability was studied by MTT assay. Insulin secretion was assessed by the enzyme-linked immunosorbent assay (ELISA). The reactive oxygen species (ROS) and cell apoptosis in normal and high glucose condition were determined by flow cytometer.
In addition, gene expression of apoptosis, antioxidant; glutathione peroxidase 1 (GPX1) and superoxide dismutase 2 (SOD2), and insulin secretory function were studied by real-time polymerase chain reaction (Real-time PCR).
**Results:** The findings revealed that FAC exposure resulted in the decrease of cell viability and insulin-release, and the induction of ROS and apoptosis in pancreatic cells. Interestingly, a combination of FAC and cytokines had an additive effect on SOD2 antioxidants' genes expression and endoplasmic reticulum (ER) stress.
In addition, it reduced the insulin secretion genes expression; insulin (INS), glucose kinase (GCK), protein convertase 1 (PSCK1), and protein convertase 2 (PSCK2). Moreover, the highest ROS and the lowest insulin secretion were found in FAC combined with IL-1β and TNF-α in the high-glucose condition of human pancreatic beta cell, which could be involved in the mechanism of DM development in β-thalassemia patients.

## INTRODUCTION

One of the hereditary blood disorders resulting from a defect in β-globin chain synthesis is β-thalassemia, which is caused by point mutation or, more rarely, deletion of the β-globin gene (*Galanello & Origa, 2010*). These defects cause an imbalance between α and β-globin chains leading to ineffective erythropoiesis (IE) chronic hemolytic anemia. Extramedullary erythropoiesis is caused by β-thalassemia, which leads to bone deformities and hepatosplenomegaly. Many complications were found such as heart failure, cirrhosis, and endocrine complications including diabetes mellitus (DM) (*Weatherall, 2003*). The prevalence of DM has been reported by up to 40% in β-thalassemia patients (*Ghergherehchi & Habibzadeh, 2015*; *Li et al., 2014*; *Liang et al., 2017*; *Metwalley & El-Saied, 2014*).

Patients with β-thalassemia and DM have a higher risk of cardiovascular complications (*Pepe et al., 2013*). However, the mechanism of DM in β-thalassemia patients is still unclear. Several mechanisms involved in DM development in β-thalassemia patients were reported including insulin resistance, hepatic dysfunction, and insulin deficiency resulting from β-cell damage or apoptosis (*Noetzli et al., 2012*; *Ghergherehchi & Habibzadeh, 2015*; *Li et al., 2014*). However, there is no report linking iron and those mechanisms in β-thalassemia patients.

Pancreatic β-cell damage is caused by many factors such as hyperglycemia, ROS, pharmacological factors, environmental toxicity factors, tumors, chronic pancreatitis, infections, inflammation, and autoimmunity. ROS caused by iron leads to damaged cells via the Fenton reaction. Iron is thought to cause cell death through ROS-dependent and -independent mechanisms (*Dixon & Stockwell, 2014*; *Bogdan et al., 2016*). Although the iron metabolism in pancreatic β-cells is still unclear, it is believed that pancreatic islet cells are susceptible to oxidative damage because they express low levels of antioxidants (*Tiedge et al., 1997*). One study of iron regulation in pancreatic islet cells reported that high expression of divalent metal transporter (DMT1) caused more iron deposits in pancreatic cells than other cells (*Andrews, 1999*).

In β-thalassemia patients, iron might not be the only factor that causes β-cell damage. Increased levels of inflammatory cytokines have been reported in β-thalassemia patients, especially TNF-α and IL-1β with mean levels 3- and 28-times higher than normal controls, respectively (*Wanachiwanawin et al., 1999*). These might cause β-cell damage through apoptosis signaling pathways in β-thalassemia patients. In addition, cytokines, especially IL-1β, also have the potential to increase intracellular iron and ROS, which cause apoptosis as has been described in insulin-producing cells (*Hansen et al., 2012*). Therefore, the increase of both iron and cytokines might be an important mechanism of DM in β-thalassemia patients. Here, the combined effect of iron and cytokines on the human pancreatic β-cell line is examined.

## MATERIALS AND METHODS

### Pancreatic cell culture

The human pancreatic β-cell line 1.1B4 was generated by electrofusion of freshly isolated human pancreatic beta cells and the human PANC-1 epithelial cell line. It showed glucose sensitivity and responsiveness to known modulators of insulin secretion (*McCluskey et al., 2011*). The 1.1B4 cells passage 28 (ECACC Cat. No. 10012801) were routinely cultured in pre-warmed RPMI640 culture medium with L-glutamine containing 11.1 mM glucose (Thermo Fisher Scientific, Waltham, MA, USA) supplemented with 10% v/v fetal bovine serum (Merck, Burlington, MA, USA) and 10 U/ml penicillin and 0.1 g/l streptomycin (Thermo Fisher Scientific, Waltham, MA, USA) at 37 °C in 5%$CO_2$.

### 1.1B4 cell treatment with FAC, cytokines and high glucose

The 1.1B4 cells were plated on six well plates with or without high glucose condition (22.2 mM glucose) (*Vasu et al., 2013*) and allowed to attach into cell plates for 24 h. Then, they were treated with different concentrations of FAC (Merck) or IL-1β or TNF-α (Cell Signaling Technology, Danvers, MA, USA) for 12 and 24 h.

### Cell viability analysis by MTT assay

MTT assay (3-(4, 5-Dimethylthiazol-2-yl)-2, 5-Diphenyltetrazolium Bromide) (Thermo Fisher Scientific, Waltham, MA, USA) was used to compare the effect of FAC, IL-1β, and TNF-α on cell viability. The 1.1B4 cells were plated on 96 well plates and allowed to attach plates for 24 h. After the treatment, the culture medium was removed, and 100 µl of culture medium was added. Then, ten microliters of 12 mM MTT was added and incubated for 4 h at 37 °C. Finally, 85 µL MTT-containing medium and 50 µl DMSO was added to each well and analyzed at 540 nm by a SpectraMax 200 (Molecular Device, San Jose, CA, USA).

### Analysis of intracellular ROS by flow cytometer

Intracellular ROS was detected by 2′,7′-Dichlorodihydrofluorescein diacetate (DCFH-DA) (Merck, Kenilworth, NJ, USA); the cells were trypsinized and washed twice with $Ca^{2+}$ and $Mg^{2+}$-free Dulbecco's phosphate buffer saline, then transferred to polypropylene (PP) tube. Then 0.05 mg/mL of DCFH-DA was added into the tube and incubated for 15 min at 37 °C, 5% $CO_2$. The fluorescence signal was detected by a fluorescence-activated cell sorting (FACS) Canto flow cytometer (BD Biosciences, San Jose, CA, USA) and was analyzed by FACSDiva software (BD Biosciences, San Jose, CA, USA).

### Analysis of apoptotic cells by flow cytometer

The apoptosis cells were detected using Annexin V: FITC Apoptosis Detection Kit I (BD Biosciences, San Jose, CA, USA). The cells were washed and incubated with fluorescein isothiocyanate (FITC) conjugated Annexin V (AnV) and propidium iodide (PI) at room temperature for 15 min. Fluorescence was detected by a FACSCanto flow cytometer and analyzed with FACSDiva software (BD Biosciences, San Jose, CA, USA).

### Gene expression by RT-qPCR

Total RNA was extracted from 1.1B4 cells using TRIzol reagent (Invitrogen, Calsbad, MA, USA). Then, the conversion of 5 μg of total RNA to cDNA using RevertAid kit (Thermo Fisher Scientific, Waltham, MA, USA) following the manufacturer's instructions, the qPCR was performed using SYBR green. The Select Master Mix for CFX (Applied Biosystem, Foster City, CA, USA) was used for RT-qPCR, through specific primers, and run on CFX96 Touch Real-Time PCR (Bio-Rad, Hercules, CA, USA). The level of target gene expression was normalized against *ATCB* expression. The mRNA fold change was calculated using the $2^{-\Delta\Delta Ct}$ method, with the values expressed as fold change relative to the untreated control.

### Insulin measurement by ELISA

The 1.1B4 cell was treated with FAC alone, cytokine (IL-1β or TNF-α alone), two cytokines (IL-1β + TNF-α), FAC combined with each cytokine (FAC + IL-1β or TNF-α), and FAC combined with both cytokines (FAC + IL-1β + TNF-α) for 24 h. Then, insulin secretion was assayed by ELISA using the human insulin ELISA kit (Merck, Kenilworth, NJ, USA), according to the manufacturer's protocol.

### Statistical analysis

The results were analyzed by one-way and two-way analysis of variance (ANOVA) and presented in mean ± SEM of three independent experiments. The difference in results was considered significant when $p < 0.05$.

## RESULTS

### FAC induce ROS production, reduce cell viability, and decrease insulin expression of pancreatic β-cell

The accumulated iron in β-cell islets might promote ROS levels and cell apoptosis leading to a decrease in insulin production. The cellular ROS levels and cytotoxic effects of iron were first examined in pancreatic beta cells. FAC (0.1–15 mM) increased ROS levels in a dose-dependent manner ($p < 0.05$) (Fig. 1A). In addition, FAC induced pancreatic cell apoptosis and reduced cell viability in a dose- and time-dependent manner (Figs. 1B–1C and 1G). The highest ROS (MFI 1,375 ± 64.26) and apoptosis (40%) were found in 15 mM FAC-treated cells at 24 h (Figs. 1A and 1B). Expression of apoptotic regulatory proteins include *BAX, BCL2* and *STAT1* were assessed to study the effect of FAC on β-cell apoptosis. In the 1.1B4 cells treated with 8 mM FAC the *BAX* were up-regulated at 12 h and down-regulated at 24 h while *BCL2* was down-regulated at 12 and 24 h (Figs. 1D–1F). The lowest cell viability (40%, $p < 0.001$) was found in cells treated with 10 and 15 mM FAC for 24 h (Fig. 1G).

Insulin expression and secretion were assessed to study the effect of FAC on β-cell function. In the 1.1B4 cells treated with 8 mM FAC the insulin (*INS*) expression and insulin level decreased in a time-dependent manner (Figs. 1H and 1I).

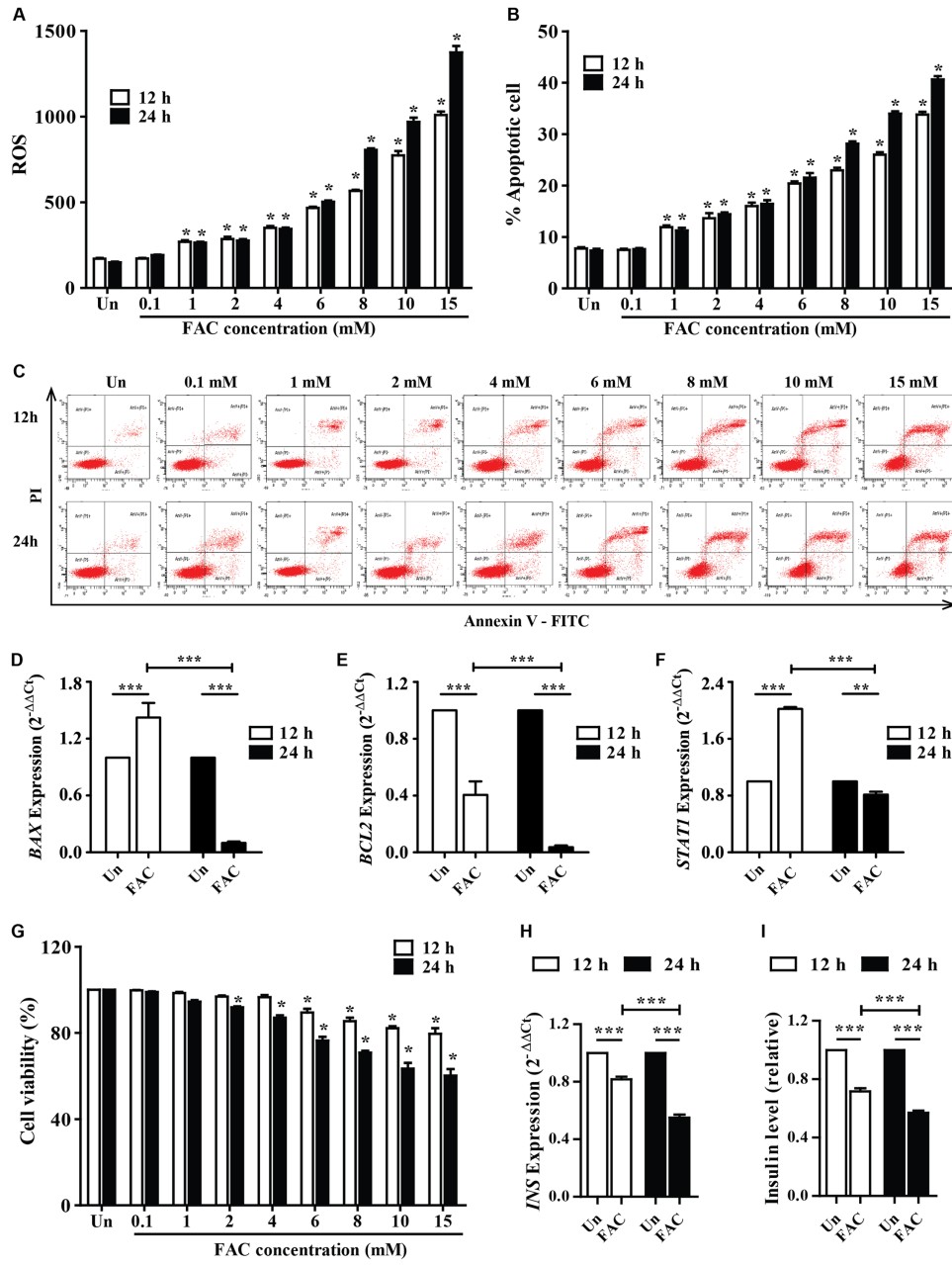

**Figure 1 Iron-induce intracellular ROS generation, increase cell death, reduce INS expression, and insulin production in pancreatic β-cells.** The 1.1B4 cells were exposed to different concentrations of FAC for 12 and 24 h. (A) ROS, cells stained with DCFH-DA were detected using a flow cytometer. (B) Apoptosis, cells were stained with fluorescein isothiocyanate (FITC)-conjugated Annexin V and PI and was detected by using a flow cytometer. (C) Flow plots of apoptosis. Apoptosis regulatory gene expression, mRNA were extracted and converted to cDNA. The mRNA expression was normalized to *ACTB* expression; (D) *BAX* expression. (E) *BCL2* expression. (F) *STAT1*: expression. (G) cell viability was measured using MTT assay. The *INS* expression was checked, and the cultured medium was collected for insulin measurement using an ELISA kit. (H) *INS* expression in $2^{-\Delta\Delta Ct}$. Later, it was normalized by *ACTB*. (I) insulin level, which was normalized by the untreated cells. The results were shown in mean ± SEM from three independent experiments and yielded. $^*p < 0.05$ and $^{**}p < 0.001$ indicate a statistically difference compared to the untreated cells. $^{***}p < 0.0001$, indicating a statistically significant difference compared with untreated cells and 12 h using a one-way ANOVA test.

## Combination of FAC and cytokines induce ROS production, apoptosis, and reduces insulin expression of pancreatic β-cells

Cytokine activates the production of ROS and reduction of insulin secretion which lead to diabetes. The effect of FAC combined with cytokines on pancreatic β-cells was observed. Cells were treated with IL-1β alone, TNF-α alone, IL-1β combined with TNF-α, FAC alone, and FAC combined with IL-1β, FAC combined with TNF-α, and FAC combined with IL-1β and TNF-α for 24 h. The results showed that ROS and percentage of apoptosis increased but had not significantly different compared between FAC with cytokines and FAC alone (Figs. 2A and 2B). Insulin secretion and *INS* expression was reduced in FAC combined with IL-1β or TNF-α compared with FAC alone (Figs. 2C and 2D).

The effect of 22.2 mM high glucose was also investigated in 1.1B4 cells, cell viability of pancreatic cell treated with cytokines with FAC was lower than in a normal glucose condition. The lowest cell viability was found in FAC combined with IL-1β and TNF-α in a high-glucose condition (Fig. 3A). In addition, the highest ROS was found in FAC combined with IL-1β and TNF-α in the high-glucose condition (Fig. 3B). Apoptosis of pancreatic cells in the high-glucose condition was higher than in the normal glucose condition. Interestingly, the highest apoptosis (Fig. 3C), lowest insulin secretion (Fig. 3D) and lowest *INS* expression were found in FAC combined with IL-1β and TNF-α in the high-glucose condition (Fig. 3E).

## FAC combined with IL-1β and TNF-α decrease secretory function genes through antioxidant defense, apoptosis, and ER stress

From these results showed that iron (FAC) and cytokines reduce insulin secretion capacity, this evidence was examined by measuring the expression of genes involved in antioxidant defense, apoptosis pathways, ER stress, and secretory function (Fig. 4). Two important antioxidant genes in the pancreas, including superoxide dismutase 2 (*SOD2*) and glutathione peroxidase 1 (*GPX.1*) were measured. The results showed that the highest *SOD2* expression was found in FAC combined with TNF-α alone and FAC combined with IL-1β and TNF-α (Fig. 4A). In addition, apoptosis-related genes, including B-cell lymphoma 2 (*BCL2*), Nuclear factor-κB (*NFκB*), and signal transducer and activator of transcription 1 (*STAT1*) were studied. The results showed that the expression of *BCL2*, *NFκB*, and *STAT1* was decreased (Figs. 4C–4E). Gene expressions of Heat Shock Protein Family A (Hsp70) Member 4 (*HSPA4*) and Heat Shock Protein Family A (Hsp70) Member 5 (*HSPSA5*), ER stress marker genes, were analyzed. The highest gene expression of *HSPA4* was shown in FAC combined with both cytokines. However, the gene expression of *HSPA5* was decreased (Figs. 4F and 4G). Moreover, insulin-secretion related genes including glucose kinase (*GCK*), proprotein convertase subtilisin/kexin type 1 (*PCSK1*), and proprotein convertase subtilisin/kexin type 2 (*PCSK2*) were determined. The expressions of these genes were decreased more than two times in FAC treated alone and in combination with cytokines (Figs. 4H–4J).

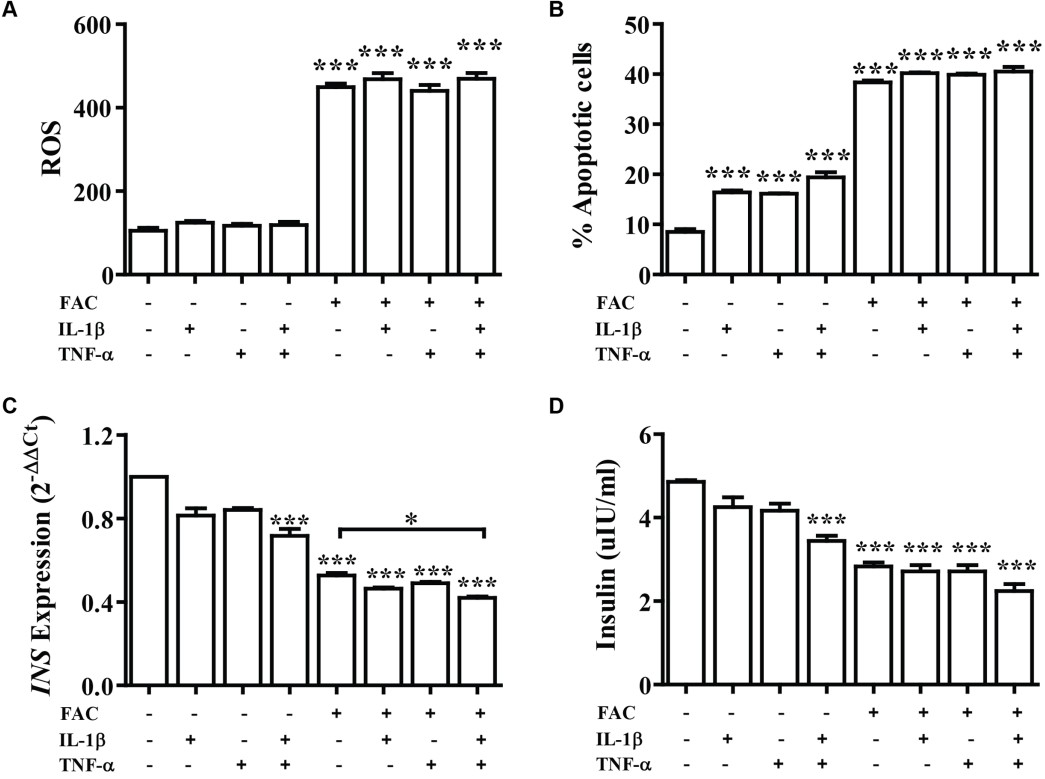

**Figure 2** **The effect of FAC combine with IL-1β and TNF-α.** The 1.1B4 cells were treated with 8 mM FAC combined with 10 pg/mL of IL-1β, FAC combined with 20 pg/mL of TNF-α, and FAC combined with IL-1β and TNF-α for 24 h. (A) ROS cells stained with DCFH-DA were detected using a flow cytometer. (B) Apoptosis cells stained with fluorescein isothiocyanate (FITC)-conjugated AnV and PI were detected using a flow cytometer. (C) *INS* expression, mRNA were extracted and converted to cDNA. The mRNA expression was normalized to *ACTB* expression. (D) Insulin level. The culture medium was collected for insulin measurement using the ELISA technique. The results are shown in mean ± SEM from three independent experiments and yield a ***$p < 0.0001$, indicating a statistically significant difference compared with the untreated cells and a *$p < 0.05$, indicating a statistically significant difference compared with FAC treated alone.

# DISCUSSION

DM is commonly found in β-thalassemia patients. The overload of iron and increased of cytokines may affect the β-cell function and insulin secretion capacity, which lead to DM (*Wanachiwanawin et al., 1999*). However, the mechanism of DM in thalassemia is still unclear. The purpose of this research is to study the effect of FAC combined with cytokines IL-1β and TNF-α on 1.1B4 in the human pancreatic islet cell line.

Firstly, we found that iron exposure resulted in a reduction of cell viability, induction of ROS, and increasing of apoptosis in a dose-dependent manner in 1.1B4 cells that have been previously reported in other types of cell line such as HH4, human hematopoietic cells/mesenchymal cells, MC3T3-E1 cells, and MG-63 cells through different mechanisms (*Li et al., 2016*; *Lu et al., 2013*; *Doyard et al., 2012*; *Yang et al., 2017*; *Ke et al., 2017*). Many reports have been demonstrated that iron overload lead to increasing intracellular ROS and caused cell damage, which is concomitant with our results. Moreover, our results showed the reduction of *INS* expression and insulin secretion that might be affected by

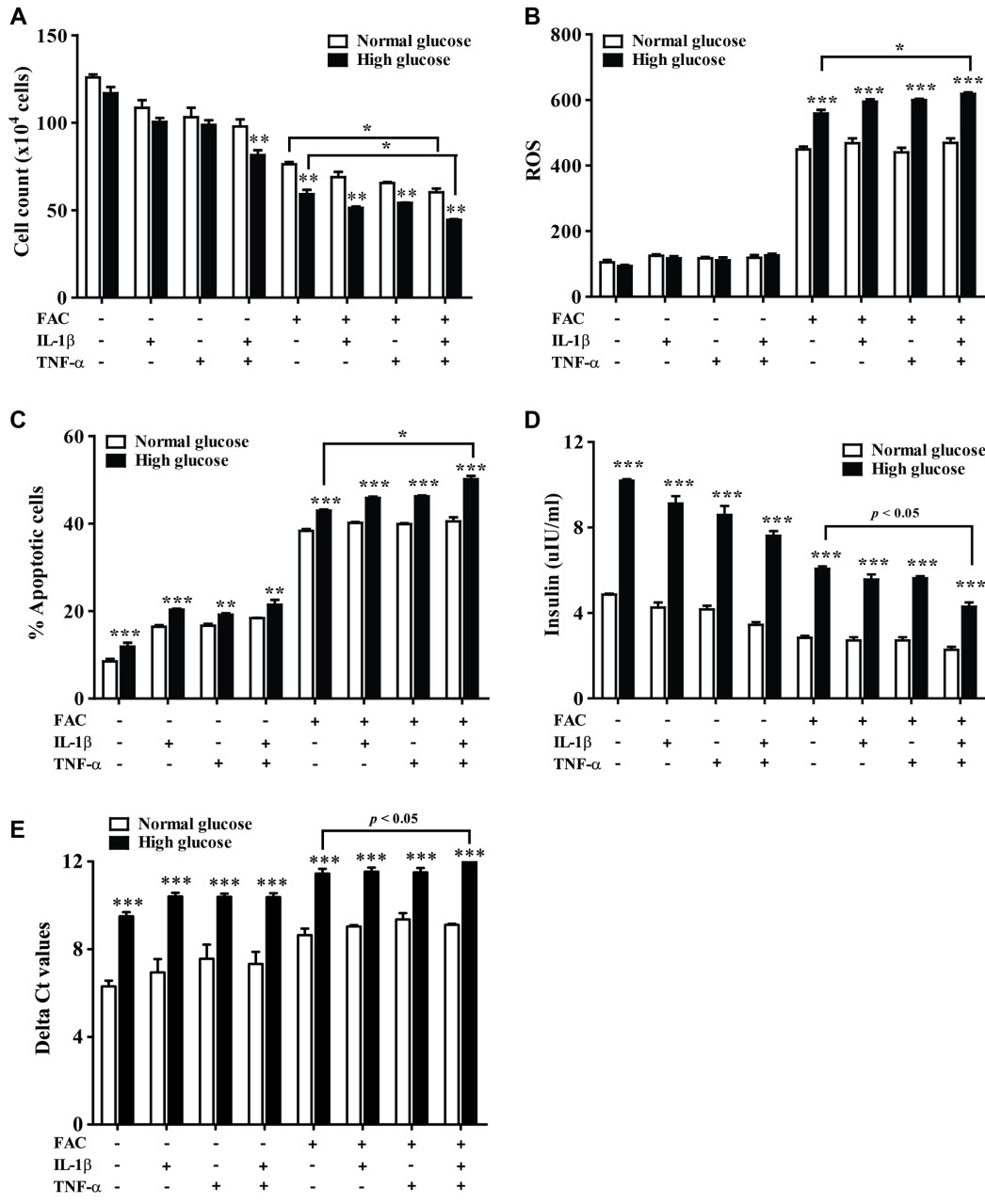

**Figure 3 The effect of high glucose on iron combined with IL-1β- and TNF-α-treated cells.** The 1.1B4 cells were treated with 8 mM FAC combined with 10 pg/mL of IL-1β, FAC combined with 20 pg/mL of TNF-α, and FAC combined with IL-1β and TNF-α in no adding glucose and 22.2 mM glucose adding condition for 24 h. (A) Cell count, the cells stained with typhan blue were counted under a microscope. (B) The ROS, cells stained with DCFH-DA were detected using a flow cytometer. (C) The apoptosis cells stained with fluorescein isothiocyanate (FITC)-conjugated AnV and PI were detected using a flow cytometer. (D) Insulin level, the culture medium was collected for insulin measurement using the ELISA technique. (E) *INS* expression, mRNA were extracted and converted to cDNA. The mRNA expression was shown as delta Ct values calculated from $Ct_{INS}$ to $Ct_{ACTB}$. The results are shown in mean ± SEM of three independent experiments and revealed a **$p < 0.01$ and a ***$p < 0.0001$, indicating the statistically significant differences compared to the no adding glucose. *represent the significantly difference at $p < 0.05$.

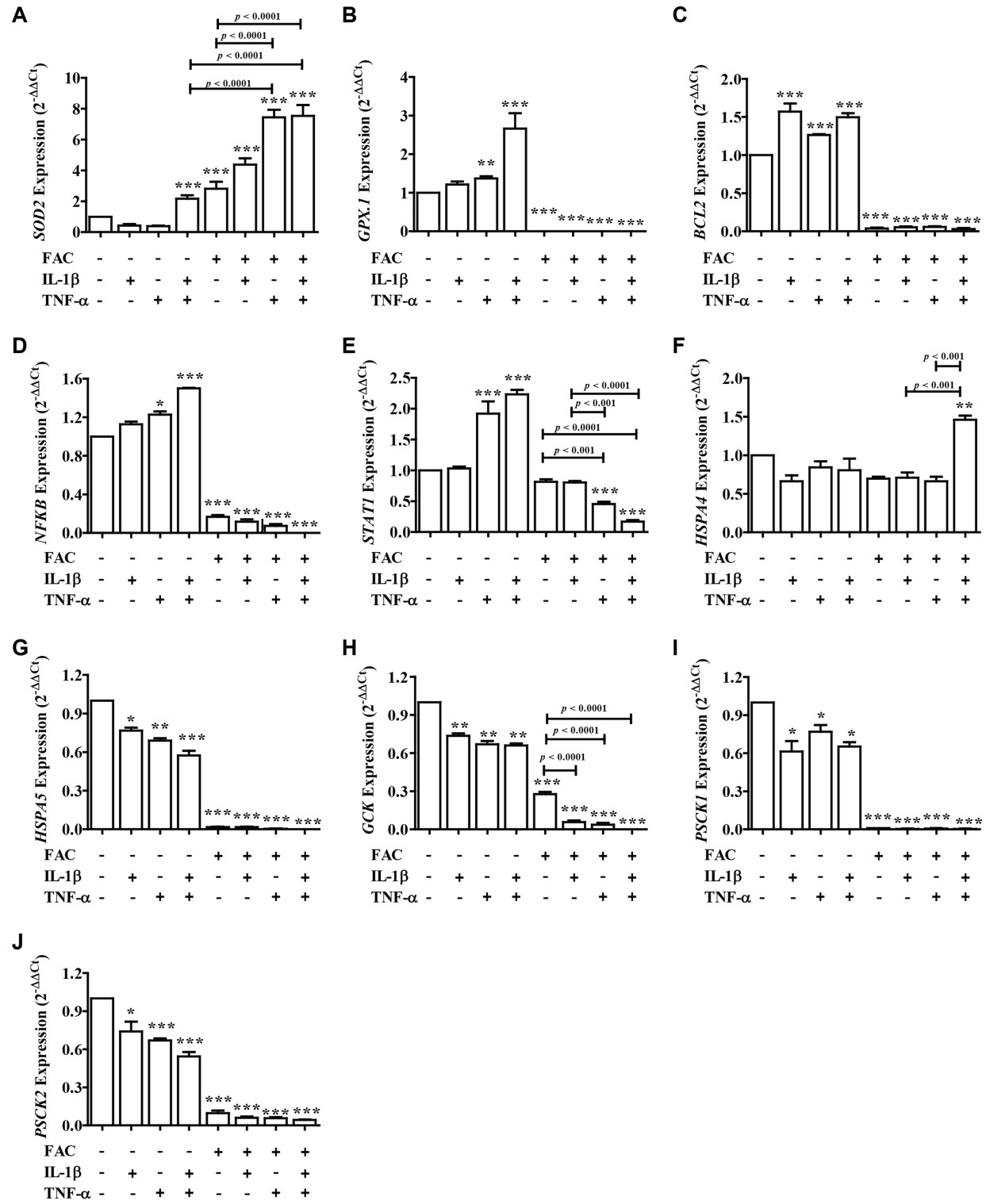

**Figure 4 The expression of genes involved in insulin secretion, antioxidants, ER stress, and apoptosis signaling pathways.** Following FAC and cytokines exposure, mRNA was extracted and converted to cDNA. The mRNA expression was normalized to *ACTB* expression. The results are shown as mean ± SEM of three independent experiments and indicated a $^*p < 0.05$, $^{**}p < 0.01$, and $^{***}p < 0.0001$, indicating the statistically significant differences compared to the untreated cells. The antioxidant genes include *SOD2* (A) and *GPX.1* (B). The apoptosis pathway genes are *BCL2* (C), *NFκB* (D), and *STAT1* (E). ER stress genes include *HSPA4* (F) and *HSPA5* (G). The secretory function genes include *GCK* (H), *PSCK1* (I), and *PSCK2* (J). 

ROS induced by FAC, which finally cause β-cell damage. FAC was previously used to induce the excess intracellular iron and caused increasing of ROS affected insulin secretion in Rat insulinoma pancreatic β-cells (RINm5F) as a model of pancreatic iron overload in β-thalassemia patients (*Koonyosying et al., 2019*).

The toxic effects of cytokines was studied but the mechanism or pathway underlying those effects in pancreatic β-cell is still unclear. Numerous studies whose findings reported that short-term treatment with cytokines such as IL-1β, TNF-α, and interferon γ (IFN-γ), alone or in combination could induce apoptosis of islet cells and dysfunction of pancreatic β-cells to release insulin (*Vasu et al., 2014*; *Wang, Guan & Yang, 2010*), are concomitant with our results.

Interestingly, β-thalassemia patients with iron overloading had high levels of these cytokines and high percentage of apoptosis. The iron and cytokines-induced apoptosis might be the cause of β-cell dysfunction and DM development in these patients. From our results, FAC combined with IL-1β and TNF-α had an additive effect on reducing *INS* expression, inducing *SOD2* antioxidants' genes expression and activating endoplasmic reticulum (ER) stress. The increasing of ROS in FAC combined with IL-1β and TNF-α induced the expression of antioxidant *SOD2* have been reported (*Wang et al., 2018*). In addition, the induction of *SOD2* expression by cytokines was reported (*Vasu et al., 2014*). Moreover, the result revealed that the *HSPA4* heat shock 70 kDa protein 4 genes responded to the activation of oxidative stress, protein-folding, and secretion demands that can be found in hyperglycemia (*Berchtold et al., 2016*). Jacob and co-workers found that 100 pg/ml of IL-1β induced divalent metal transporter 1 (DMT1) expression and caused increasing of ROS and activating apoptosis pathway in pancreatic islets cells (*Hansen et al., 2012*). This information is not consistent with our results that ROS and apoptosis in FAC combined with IL-1β and/or TNF-α were not significantly different from FAC treated alone.

Iron-induced ROS can activate ferroptosis which might be the possible mechanism of pancreatic β-cell damage in our study. It is a non-apoptosis cell death is induced by the small molecule or condition that inhibits glutathione biosynthesis or the cellular glutathione-dependent antioxidant enzyme glutathione peroxidase 4 (GPX4) and results in an accumulation of ROS (*Cao & Dixon, 2016*; *Xie et al., 2016*).

Experiments conducted with high glucose concentrations shows significant apoptotic death compared to normal medium in untreated cells however no change in cell count and ROS, because high glucose could activate apoptosis signaling pathway and down-regulation of anti-apoptotic proteins *BCL2*, *MAPK8*, and *MAPK10* (*Vasu et al., 2013*). However, only high glucose had less affect to cell count and level of ROS but when combine high glucose with FAC, and cytokines could lead to increase level of ROS. Hyperglycemic condition in the combination of FAC and cytokines induce production of ROS and lead to apoptosis and cause reducing of insulin in 1.1B4 cells that might be resulting from the down regulation of *INS* and other genes involved in insulin secretion, including *GCK*, *PSCK1*, and *PSCK2* (*Vasu et al., 2014*; *Kooptiwut et al., 2005*; *Jonas et al., 2009*; *Laybutt et al., 2002*). This might be due to the interference of high glucose that reduces the *PDX1*, an important gene involved in the secretory function (*Sachdeva et al., 2009*). The effect of cytokines on insulin production in hyperglycemia was found

(*Zhang & Kim, 1995*; *Kiely et al., 2007*). Moreover, it was found that glucose toxicity induced ROS production and activated the apoptosis signaling pathway (*Robertson, 2004*). This information is concomitant with our study that the highest ROS, the lowest *INS* expression, and insulin secretion were found in FAC combined with IL-1β and TNF-α in high glucose level. This results could be supported the mechanism of insulin dependent or DM development in β-thalassemia patients who had iron overload and high level of cytokines IL-1β or TNF-α.

However, this study was performed only in human pancreatic β-cell line as a model which no direct measurement in humans and external validation but our previous data suggests that in β-thalassemia/HbE patients with diabetes had high level of ROS, low level of glutathione, reduced the β-cell function (HOMAB) and insulin level decreased (P. Rattanaporn, 2019, unpublished data).

## CONCLUSIONS

This study suggests that FAC combined with cytokines, IL-1β and TNF-α in high glucose condition could increase ROS production, ER stress and apoptosis induction. However, they reduce insulin secretion in human pancreatic β-cell lines. This information could be benefit for understanding the mechanism on development of DM in thalassemia patient.

## ACKNOWLEDGEMENTS

We gratefully acknowledge Thalassemia Research Center, Institute of Molecular Bioscience and Faculty of Medical Technology, Mahidol University, Thailand.

### Funding

This study was supported by Mahidol University Research Grants, Research Chair Grant, NSTDA (NSTD), and the Office of the Higher Education Commission and Mahidol University under the National Research University Initiative. Patchara Rattanaporn was supported by the Thailand Graduate Institute of Science and Technology (TGIST), NSTDA (grant number TG221457037D). The funders had no role in study design, data collection and analysis, decision to publish, or preparation of the manuscript.

### Grant Disclosures

The following grant information was disclosed by the authors:
Mahidol University Research Grants, Research Chair Grant, NSTDA (NSTD).
National Research University Initiative.
Thailand Graduate Institute of Science and Technology (TGIST), NSTDA: TG221457037D.

### Competing Interests

The authors declare that they have no competing interests.

## Author Contributions

- Patchara Rattanaporn conceived and designed the experiments, performed the experiments, analyzed the data, prepared figures and/or tables, authored or reviewed drafts of the paper, and approved the final draft.
- Sissades Tongsima analyzed the data, authored or reviewed drafts of the paper, and approved the final draft.
- Thomas Mandrup-Poulsen analyzed the data, authored or reviewed drafts of the paper, and approved the final draft.
- Saovaros Svasti conceived and designed the experiments, authored or reviewed drafts of the paper, and approved the final draft.
- Dalina Tanyong conceived and designed the experiments, authored or reviewed drafts of the paper, and approved the final draft.

## Data Availability

The raw data are available in the Supplemental Files.

## Supplemental Information

Supplemental information for this article can be found online at http://dx.doi.org/10.7717/peerj.9298#supplemental-information.

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
