# Peer review of "Combination of ferric ammonium citrate with cytokines involved in apoptosis and insulin secretion of human pancreatic beta cells related to diabetes in thalassemia"

_PeerJ, doi:10.7717/peerj.9298_

## Round 0.1 · original submission · Minor Revisions

As you will see, both reviewers were positive regarding your study and have indicated that it makes an important contribution. Some attention and extra experimental work may be required however. In particular, some analysis of apoptosis is required, such as the analysis of expression of apoptotic regulators, and the proposal that ferric ammonium citrate drives ROS expression should be tested by using ROS inhibitors.

There is also a suggestion to test this model in another cell line which you should reflect on.

I realize that this requires extra experimental work and I am not certain of how you are equipped to deal with this during the present pandemic. I would urge you to discuss re-submission deadlines with the office who will be sympathetic.

Thank you for submitting this interesting study to us and I hope you and your colleagues are well.

Reviewer 1 ·

Basic reporting

The study was well designed and performed meeting the criteria of the journal.

Experimental design

This a nice designed and performed study using quite difficult methodologies.
No external validation of the method was performed.

Validity of the findings

The study was well designed and performed but as no external validation was performed the validity of the findings is only on researchers responsibility.

Additional comments

In this study Rattanaporn et al. aimed to investigate the effect of the interaction of FAC on cytokines, IL-1β and TNF-α, in 1.1B4 human pancreatic β-cell line. The authors add new experimental data in the pathophysiology of DM development in patients with Thalasemia. As no direct measurements in humans as well as no external validation was performed, the discussion-conclusion (last two paragraphs) might be editing in this context.

Reviewer 2 ·

Basic reporting

Iron overload may lead to dysfunction of pancreatic β-cell function and results in Insulin deficiency or resistance in β-thalassemia patients. Earlier studies have also investigated that elevated cytokines in pancreatic β-cells may damage the cellular integrity. In this manuscript, author have investigated iron supplementation together with cytokines to human pancreatic cell line and analysis of cellular anti-oxidant signaling and apoptotic cell death. This manuscript is well conceptualized and written, studies conducted with high glucose medium provides a physiological DM condition to test the hypotheses.

Experimental design

Authors need to conduct some additional experiments to improve the outcomes of this study. Some examples where the authors could provide more insight includes-
1. Figure 1- Given that FAC mediated loss of cell viability leading to apoptotic death, it is important to include the expression of some apoptotic regulatory proteins to validate the data.
2. Authors should include the flow plots for apoptotic death.
3. To support the fact that FAC treatment leading to ROS elevation, authors need to conduct experiments with ROS quencher (N acetyl cysteine-NAC) and correlate the specificity.
4. Figure 2- It is not clear in the text about interpretation for no effect of TNF and IL-1b with FAC treatment has no effect on any readouts. In contrast with high glucose (Figure 3) similar treatment has significant effect?
5. Figure 3- What is the levels of Insulin mRNA with FAC alone treated cells incubated in high glucose medium?

Validity of the findings

Authors should test some of the experiments with more than one cell line to propose a broad effect of FAC. Experiments conducted with high glucose concentrations shows significant apoptotic death compared to normal medium in untreated cells but no change in cell count and ROS? Authors should explain these rationals in the text.

Additional comments

This study will highlight the combinatorial effect of iron overload with cytokine elevation during pathogenesis of DM associated thalassemia. Still authors need to add some more data that will further enhance the outcomes of this manuscript.

---

## Round 0.2 · Minor Revisions

Thanks for addressing the issues raised. I believe this study is almost acceptable for publication. Note that some elements of English need attention, I enclose here a marked up manuscript in which I have offered some suggestions for the Abstract and Introduction - please make similar alterations throughout. I hope you find these of use - they are intended to help clarity of message.

---

## Round 0.3 · accepted · Accept

Thanks for the revisions; congratulations on your manuscript acceptance.